# Machine Learning for Surrogate Groundwater Modelling of a Small Carbonate Island

Karl Payne [1,*], Peter Chami [2], Ivanna Odle [1], David Oscar Yawson [1], Jaime Paul [3], Anuradha Maharaj-Jagdip [4] and Adrian Cashman [5]

1. Centre for Resource Management and Environmental Studies, Cave Hill Campus, The University of the West Indies, Bridgetown BB11000, Barbados
2. Faculty of Science and Technology, Cave Hill Campus, The University of the West Indies, Bridgetown BB11000, Barbados
3. Barbados Water Authority, The Pine, St. Michael, Bridgetown BB11000, Barbados
4. Independent Hydrological Consultant, Port of Spain 150123, Trinidad and Tobago
5. Akwatix Water Resources Management, Ashford, St. John BB20000, Barbados
* Correspondence: karl.payne@cavehill.uwi.edu

**Abstract:** Barbados is heavily reliant on groundwater resources for its potable water supply, with over 80% of the island's water sourced from aquifers. The ability to meet demand will become even more challenging due to the continuing climate crisis. The consequences of climate change within the Caribbean region include sea level rise, as well as hydrometeorological effects such as increased rainfall intensity, and declines in average annual rainfall. Scientifically sound approaches are becoming increasingly important to understand projected changes in supply and demand while concurrently minimizing deleterious impacts on the island's aquifers. Therefore, the objective of this paper is to develop a physics-based groundwater model and surrogate models using machine learning (ML), which provide decision support to assist with groundwater resources management in Barbados. Results from the study show that a single continuum conceptualization is adequate for representing the island's hydrogeology as demonstrated by a root mean squared error and mean absolute error of 2.7 m and 2.08 m between the model and observed steady-state hydraulic head. In addition, we show that data-driven surrogates using deep neural networks, elastic networks, and generative adversarial networks are capable of approximating the physics-based model with a high degree of accuracy as shown by R-squared values of 0.96, 0.95, and 0.95, respectively. The framework and tools developed are a critical step towards a digital twin that provides stakeholders with a quantitative tool for optimal management of groundwater under a changing climate in Barbados. These outputs will provide sound evidence-based solutions to aid long-term economic and social development on the island.

**Keywords:** deep neural networks; elastic networks; Barbados; climate-water nexus; groundwater modelling; FEFLOW; generative adversarial networks

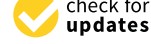



## 1. Introduction

Barbados is a small island developing state (SIDS) situated in the southeastern region of the Caribbean. The island is almost entirely dependent on groundwater for its potable water supply, with approximately 82% of its supply sourced from aquifers [1]. The remaining 18% of the island's water is supplied from desalination of brackish water. The available renewable water resources are estimated at 279 m$^3$ per person per year; a figure of less than 500 m$^3$ per person per year places Barbados in the category of absolute water scarcity according to the Falkenmark index [2]. The geology of the island is primarily a Pleistocene coral rock, which covers 85% of the island [3]. The unconfined limestone aquifer is underlain by an aquitard, which is composed of Tertiary-age units. The karstic and unconfined nature of the island's aquifers makes it vulnerable to various water quality threats. Moreover,

changes in land use such as an increase in impervious surfaces reduce the potential for aquifer recharge.

To date, the hydrogeologic studies conducted for the island have primarily been field investigations commissioned by the Barbados Water Authority (BWA), the local water utility. An early study performed by the British Union Oil Company Limited estimated that of the average annual rainfall, 20–30% recharges the island's aquifers. The total abstraction rate from the island's aquifers in 1947 was estimated to be 19,100 m$^3$/day [4]. The major findings from a subsequent study aimed at an improved understanding of Barbados' water quantity and quality estimated a safe yield of 136,000 m$^3$/day, based on an average annual rainfall value of 1500 mm, and estimated water demand of 60,000 m$^3$/day [5]. Isotopic hydrology studies were performed to investigate the spatio-temporal dynamics of recharge mechanisms on the island [3,6,7]. The results from these studies showed that the occurrence of recharge to the limestone aquifer takes place through the mechanisms of diffuse or discrete recharge and primarily occurs within the three wettest months of the year. Diffuse recharge occurs through soils comprising the vadose zone, whereas discrete recharge dominates in areas with high sinkhole densities and dry valleys. Oxygen isotopic analyses found that recharge varies between 15–20% of average annual rainfall in areas of higher elevation, increasing to 25–30% at lower elevations [3,6,7].

While there have been several field hydrogeology investigations for the island, there has been a dearth of quantitative studies. Numerical groundwater models are powerful quantitative tools that have been leveraged to quantify the response of aquifers to climatic and hydrologic stresses [8–10]. An accurate representation of Barbados' hydrogeology using numerical modelling and data-driven approaches would represent a significant advancement that would allow for more scientifically sound decision making.

Various numerical modelling approaches have been used to quantify regional flow and understand groundwater dynamics in karst and island settings similar to Barbados [11–14]. However, numerical models can be computationally expensive, which becomes problematic in groundwater resources applications where simulation-optimization is used for optimal management strategy development [15–20]. Surrogate models are a promising technique that overcome the aforementioned difficulty. They are empirical techniques that approximate a groundwater model, which describe the input–output mapping of the original model [21]. The prediction of groundwater levels has been one of the applications in several case studies where artificial intelligence has been successfully utilized [22–24] across diverse study areas. However, there is a relative dearth of applications in a small island developing state (SIDS) context.

In the taxonomy of surrogate models there are three approaches: data-driven surrogates, projection-based methods, and multifidelity-based surrogates [21]. In this paper, we implement data-driven surrogates using deep neural networks (DNNs), elastic nets, and generative adversarial networks (GANs) [25–27]. Moreover, an advantage of using DNNs is that decision support is critical for the BWA, where short runtimes are required to enable operational capabilities.

Therefore, the overall goal of this study is two-fold: (1) the development of a numerical groundwater using the FEFLOW software that accurately represents Barbados' hydrogeology, and (2) the implementation of data-driven surrogates that utilize machine learning to approximate the numerical groundwater model of Barbados. Collectively, these findings will improve the ability to manage water resources in a water scarce small island developing state (SIDS) under a changing climate.

## 2. Materials and Methods

### 2.1. Study Area Geology and Hydrogeology

Barbados (13°100′ N, 59°300′ W) is the easternmost island in the eastern Caribbean, with a total land area of 430 km$^2$ (Figure 1). Approximately 85% of the island is comprised of Pleistocene limestone geologic units that constitute the island's aquifers. These geologic formations are up to 100 m thick and in response to a continuous process of uplift, coral

reefs developed outward from the center of the island, forming terraces. The First and Second High Cliffs separate the three main groups of terraces. The Second High Cliff is roughly 30 m high and there is a high degree of karstification adjacent to this cliff [6,7]. The remaining 15% of the island's area, the Scotland District, is comprised of the Oceanic Group, where low-permeability sediments are exposed at the surface. Since surface runoff dominates in this portion of the island, the only significant source of groundwater occurs within the Pleistocene limestone formations. The island's geology has typical karst features including caves, springs, and sinkholes.

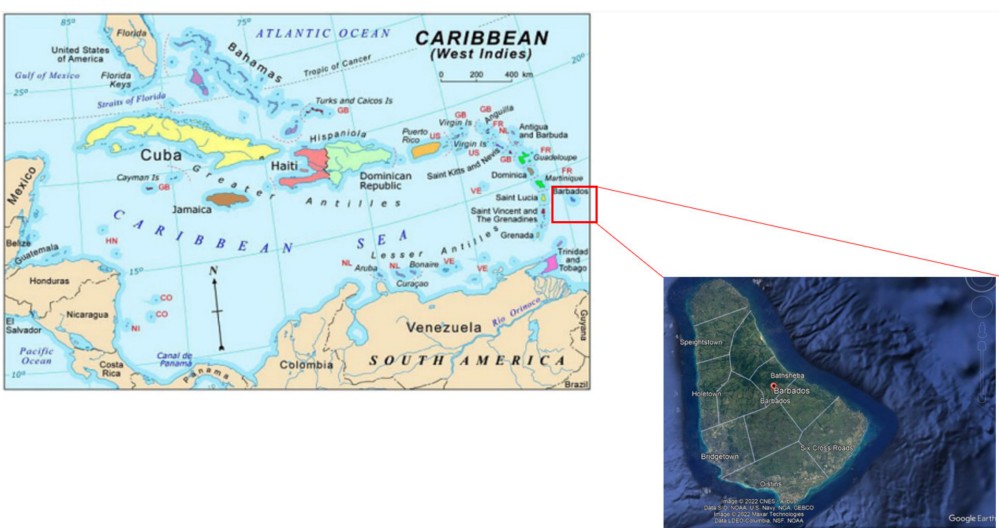

**Figure 1.** Map of the Caribbean Region showing the location of Barbados [Source: Google Earth].

The topography of the interface between the limestone and underlying sediments influences the occurrence of groundwater. Where the interface lies below mean sea level (MSL), groundwater occurs as lenses of fresh water underlain by salt water. Conversely, in locations where the aquifer–aquitard interface lies above MSL, groundwater exists as a stream [28]. Generally, the subsurface has not been well characterized across the island, therefore the interconnectivity of karst features is unknown. Therefore, the influence of preferential flow paths enhanced by limestone dissolution and local-scale flow dynamics is difficult to determine.

### 2.2. Physics-Based Numerical Model Development Using FEFLOW

#### 2.2.1. Groundwater Model Setup

One of the main assumptions of the conceptual model is that the aquifer can be represented as a single continuum. This assumption does not account for the fluxes between the limestone matrix and karst conduit network. It has been shown previously that this equivalent continuum model conceptualization is adequate for describing karst systems if the purpose of the model is to simulate regional groundwater flow [11]. The following are further assumptions made during development of the conceptual model:

1. Flow is three-dimensional and the unconfined limestone aquifer can be represented as a single continuum.
2. The underlying aquitard has a sufficiently low permeability to ignore fluxes across the interface between the base of the limestone and underlying aquitard.
3. The entire model domain is completely saturated and there are no variable density considerations.
4. The northeastern portion (Scotland District) of the island is comprised of impervious geologic formations and the permeability of these lithologies is negligible.

Boundary conditions are a mathematical statement assigning a value of the dependent variable (hydraulic head) or derivative of the dependent variable (groundwater flux) at the boundaries of the model domain. Since hydraulic heads are generally easier to measure relative to groundwater flux, it is expedient to use hydraulic heads for boundary condition specification [29]. Therefore, the coastline was assigned a Dirichlet boundary condition to account for a specified head of 0 m indicating mean sea level. The model area considered was only the Pleistocene limestone (approximately 365 km$^2$); the remainder of the island was assigned inactive cells. The model was run to a steady-state and accounted for fully three-dimensional groundwater flow. The model domain was discretized using a finite element mesh with over 2,000,000 elements. The vertical discretization was 25 layers required to approximate the limestone aquifer only.

### 2.2.2. Physics-Based Groundwater Model Calibration

The island was delineated into four different zones using a sinkhole map as a proxy for degree of karstification. Figure 2 (left) shows the spatial distribution of sinkholes across the island, while Figure 2 (right) shows the sinkhole zones. These karstic sinkholes are collapse features induced by dissolution of the island's limestone geology. Table 1 provides a description of each of the four sinkhole zones (SZ1–SZ4) along with their corresponding hydraulic conductivity parameterization. Areas with a higher density of sinkholes were assumed to have a higher degree of karstification, thereby allowing for more rapid groundwater flow. Conversely, areas with a lower sinkhole density were assumed to transmit groundwater more slowly, therefore a smaller hydraulic conductivity value was assigned. Based on experimental evidence from isotope hydrology studies, an average annual recharge value of 300 mm/year was assigned to the top of the model domain.

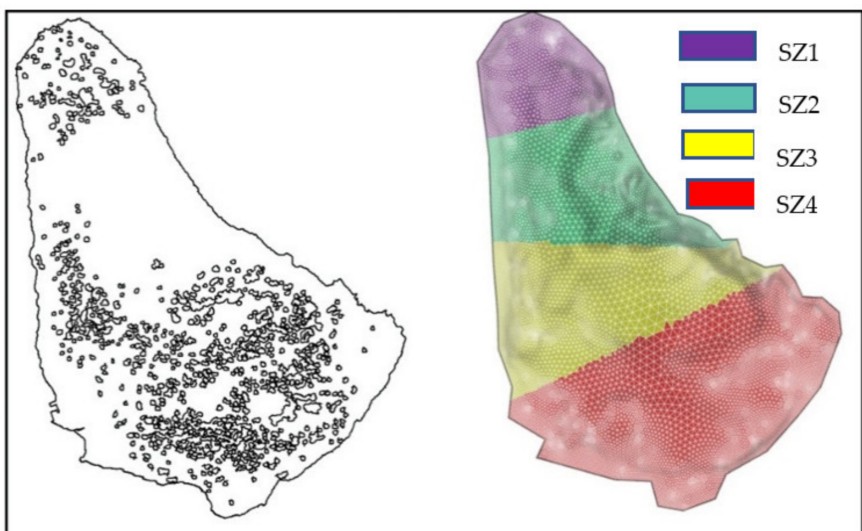

**Figure 2.** Sinkhole map showing spatial variation in sinkhole density (**left**). Delineation of four sinkhole zones with each being assigned different hydraulic conductivities (**right**).

**Table 1.** Specification of sinkhole zones and initial parameterization.

| Sinkhole Zone | Description | K (m/day) |
|---|---|---|
| SZ 1 | Medium-density zone | 426 |
| SZ 2 | Low-density zone | 10 |
| SZ 3 | High-density zone | 1500 |
| SZ 4 | High-density zone | 1500 |

The model was run to a steady-state condition and the calibration process was automated using FePest, which uses a Gauss–Newton-Levenberg–Marquardt algorithm to

minimize the difference between simulated and observed hydraulic heads [30]. The hydraulic conductivity field was assumed to be homogeneous and isotropic within each of the four zones. The BWA measures the depth to the water table at boreholes across the island. These data were transformed into hydraulic head values by deducting the depth to water from the surface elevation. The surface elevation data were obtained from a digital elevation model (DEM) from the Barbados Lands and Surveys Department.

### 2.3. Machine Learning for Surrogate Groundwater Modelling

Groundwater level prediction in response to climatic and hydrologic stresses is critical for sustainable groundwater resources management. Fluctuations in hydraulic head in aquifers affect sea water intrusion in coastal zones, ecosystem services, and soil stability. Therefore, it is of the utmost importance to have a quantitative understanding of the response of groundwater systems to pumping and recharge. Assuming fully saturated conditions, the spatio-temporal dynamics of the hydraulic head distribution is predicted by the following equation subject to the appropriate initial conditions (*I.Cs*) and boundary conditions (*B.Cs*):

$$\frac{\partial}{\partial x}\left(K_{xx}\frac{\partial h}{\partial x}\right) + \frac{\partial}{\partial y}\left(K_{yy}\frac{\partial h}{\partial y}\right) + \frac{\partial}{\partial z}\left(K_{zz}\frac{\partial h}{\partial z}\right) \pm Q_i = S_s\frac{\partial h}{\partial t} \tag{1}$$

$$h = h_0 \, , \; t = t_0 \; \; (I.C) \tag{2}$$

$$h = 0 \; \; on \; \; \Omega_c \; (B.C) \tag{3}$$

where $h(x,y,z)$ is hydraulic head, $h_0$ is the initial hydraulic head, $K_{xx}$, $K_{yy}$, and $K_{zz}$ are the hydraulic conductivity values in the $x$, $y$, and $z$ directions, $Q_i$ is the source/sink term, $S_s$ is specific storage, $t$ is time, $t_0$ is the initial simulation time, and $\Omega_C$ is the coastal boundary. The FEFLOW software also referred to as the physics-based model was used to solve the system of Equations (1)–(3) to determine the hydraulic head distribution across the model domain.

Three different machine learning techniques were used to construct surrogates for the physics-based groundwater model. The training and testing data were generated by computational experiments, which consisted of running 500 simulations of the calibrated FEFLOW model with various pumping rates in a range of 3400 m$^3$/day to 10,300 m$^3$/day at 5 different locations (see Figure 3). This range was chosen to reflect a plausible range of worst to best case scenarios for groundwater abstraction, given that total groundwater abstraction rates are approximately 70 million m$^3$/year and there are 22 wells used for abstraction by the BWA. The recharge rate was varied in the range of 70 mm/year to 1050 mm/year to capture a range from drought to above average annual rainfall conditions. The time horizon considered was one year and the objective was to learn the input–output relationship between hydrologic stresses (pumping and recharge rates) and the hydraulic head at the end of the stress period.

Figure 3 is a flowchart showing the methodology used to meet the objective of developing an accurate ML surrogate for steady-state hydraulic head prediction. There are broadly four steps in the process to meet the objective of approximating the physics-based steady-state heads using ML techniques.

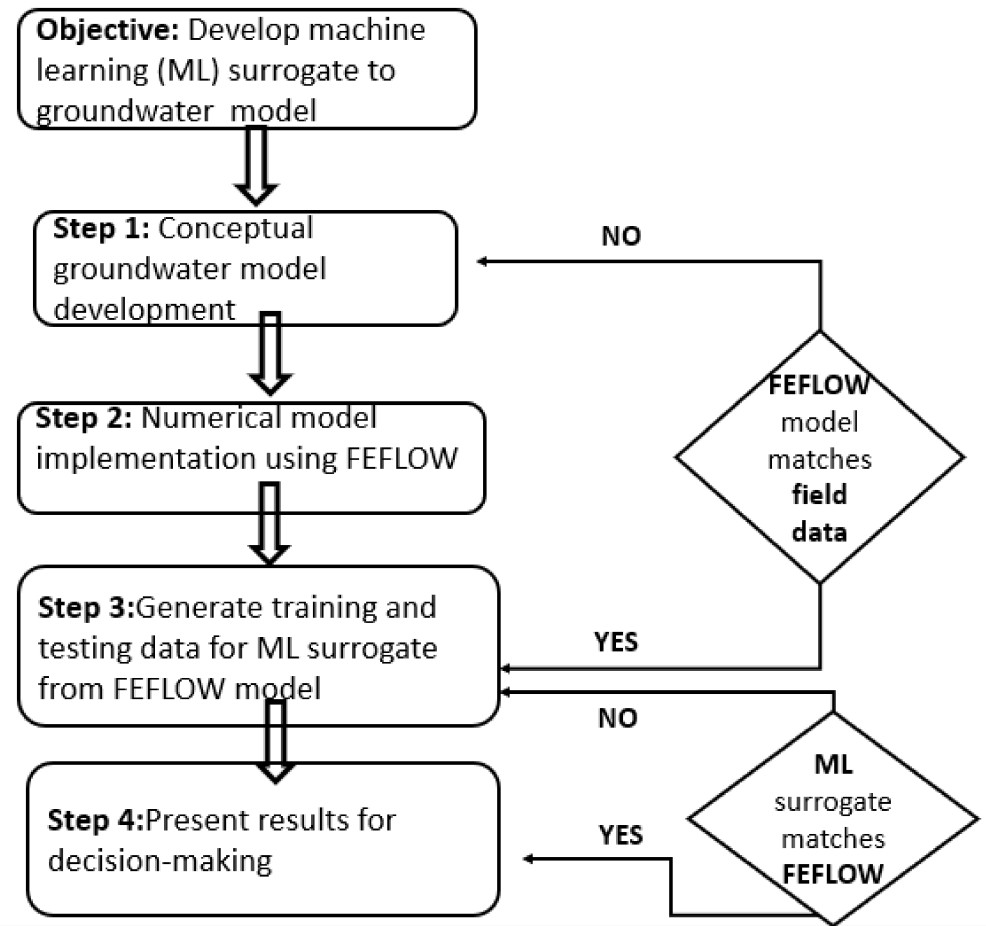

**Figure 3.** Flowchart showing the methodology used to meet the objective of developing an accurate ML surrogate for steady-state hydraulic head prediction.

2.3.1. Deep Neural Networks for Surrogate Modelling

Deep neural networks (*DNN*s) consist of multiple layers of artificial neurons, which receive an input and perform a non-linear transformation. Mathematically, a feedforward DNN used in supervised learning is defined as follows:

Let $DNN^K(x)$: $\mathbb{R}^c \rightarrow \mathbb{R}^s$ be a K-layer neural network with $N_k$ neurons in the *k*-th layer. Further, denote the bias vector and weight matrix in the *k*-th layer by $b^k$ and $W^k$, respectively. For a non-linear activation function, which is typically a rectified linear unit (ReLU) given by max {*x*, 0}, the sequential processing of a DNN is:

$$input\ layer:\ DNN^0(x) = x \in \mathbb{R}^c \tag{4}$$

$$hidden\ layers:\ DNN^k(x) = \sigma\left(W^k DNN^{k-1}(x) + b^k\right) \tag{5}$$

$$output\ layer:\ DNN^k(x) = W^k DNN^{k-1}(x) + b^k \tag{6}$$

Figure 4 shows the structure of the feedforward *DNN* defined in mathematical terms in Equation (4). The input features are well abstraction rates at 5 different pumping wells (PW-1–PW-5) and a recharge rate, while the output feature is the stead-state hydraulic head at an observation well. The *DNN*s were implemented using the Python library Keras with a TensorFlow backend.

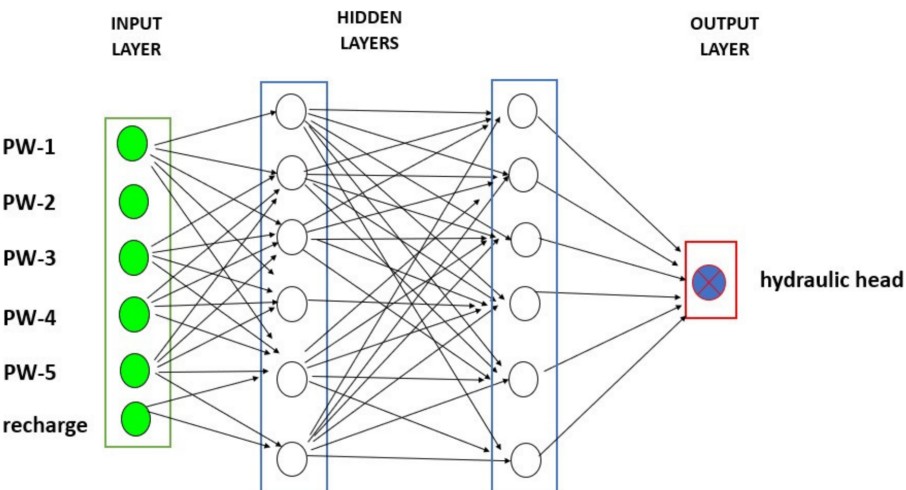

**Figure 4.** Deep feedforward neural network architecture used for learning the mapping between hydrologic stresses (pumping and recharge) and hydraulic head.

The 500 data points generated from the physics-based groundwater model were split into 400 training samples and 100 testing samples. The sampling was performed with the Scikit-learn Python library using a pseudo-random train-test split to minimize the introduction of bias in the analysis. Figure 5 shows the location of pumping wells where the abstraction rates were varied as well as the location of the observation well where the hydraulic head was predicted in response to varying pumping and recharge rates.

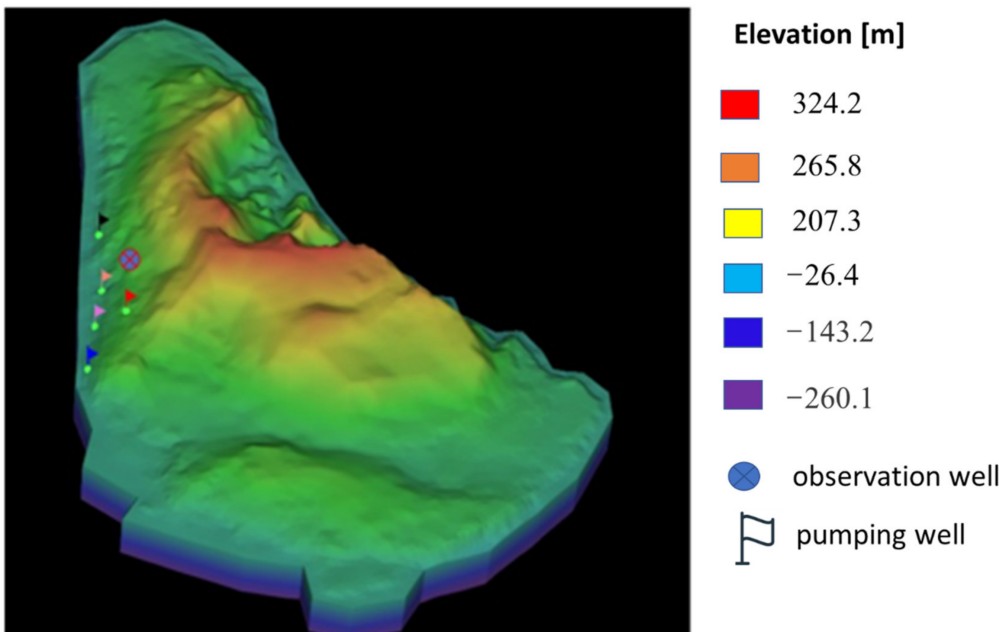

**Figure 5.** Map showing a 3D representation of the model domain with the location of pumping and observation wells used for the surrogate modelling.

### 2.3.2. Elastic-Net Regression for Surrogate Modelling

The elastic net first emerged as a result of critique on the original regularization regression method least absolute shrinkage and selection operator regression, or more commonly known as lasso, whose variable selection can be too dependent on data and thus unstable [26]. The solution to the perceived shortcoming of the lasso method is to combine the penalty methodology of the second type of regularization; ridge regression and lasso to obtain benefits from both techniques. Elastic nets aim to minimize the following loss function:

$$L_{enet}(\hat{\beta}) = \frac{\sum_{i=1}^{n}(\hat{y} - x_i'\hat{\beta})^2}{2n} + \lambda\left(\frac{1-\alpha}{2}\sum_{j=1}^{m}\hat{\beta}_i^2 + \alpha\sum_{j=1}^{m}|\hat{\beta}_j|\right) \tag{7}$$

where $m$ is the number of predictor variables, $n$ is the number of data points, $L_{enet}(\hat{\beta})$ is the elastic net loss function, $\alpha$ is the mixing parameter between ridge ($\alpha = 0$) and lasso ($\alpha = 1$), $\lambda$ and $\alpha$ are tuning parameters. By iteratively updating the tuning parameters, the elastic net learns the mapping between input features and output.

The strength of elastic net relative to other machine learning methods is that it performs variable selection and regularization. This is especially useful when there are highly correlated input features. A penalty was imposed using a shrinkage regression approach to penalize non-parsimonious models. To avoid over-fitting, cross-validation using a 10-fold with 10 repeats was employed. The elastic net regression was implemented using the R software for statistical computing.

### 2.3.3. Generative Adversarial Neural Networks (GANs) for Surrogate Modelling

Generative adversarial networks (GANs) are comprised of two models: a generative model, *G*, and a discriminative model, *D*. The role of the generator is to create data in such a way that it can deceive the discriminator, while the role of the discriminator is to distinguish between actual and generated data [27]. Mathematically, the problem is posed as a two-player zero-sum game in which the loss function for an entire data set is defined as:

$$min_G max_D V(D,G) = min_G max_D \left(E_{xP_{data}(x)}[(x)] + E_{zP_z(z)}[log(1 - D(G(z))\right) \tag{8}$$

and the individual loss functions for the generator and discriminator are:

$$L^G = min[log(D(x)) + log(1 - D(G(z)))] \tag{9}$$

$$L^D = max[log(D(x)) + log(1 - D(G(z)))] \tag{10}$$

where *D(x)* is the discriminator's evaluation of real data, *D(G(z))* is the discriminator's evaluation of fake data, *G(z)* is the generator's fake data, $L^D$ is the discriminator loss, $L^G$ is the generator loss, $P_{data}(x)$ is the original data distribution, $P_g(x)$ is the generated distribution, $P_z(z)$ is the input noise distribution, and *V(D,G)* is the expectation of the combined loss function.

The objective of the GANs developed was to generate data that are indistinguishable from the FEFLOW physics-based simulations. The GANs were implement using Keras with a TensorFlow backend. The output variable (hydraulic head) was split into bins using the minimum and maximum values of the variable to create these bins. The data set was then transformed to obtain a Gaussian distribution using a power transformer. The binary cross-entropy loss function was utilized, which is given by the following:

$$H_p(q) = -\frac{1}{N}\sum_{i=1}^{N} y_i.log(p(y_i)) + (1 - y_i).log(1 - p(y_i)) \tag{11}$$

## 3. Results and Discussion

### 3.1. Physics-Based Simulations of Island-Scale Groundwater Flow

There was an excellent agreement between the FEFLOW model and observed hydraulic head. Figure 6 shows a plot of the predicted versus field measured hydraulic head data. The root mean squared error (RMSE) and mean absolute error (MAE) were 2.7 m and 2.08 m, respectively, and there was no noticeable bias in the FEFLOW model to overestimate or underestimate the data. The hydraulic conductivity field required to describe the data for SZ1, SZ2, SZ3, and SZ4 were 400 m/day, 10 m/day, 1500 m/day, and 1500 m/day, respectively. These values are within a reasonable range for limestone and karstic limestone geology, which ranges from 0.09 to 1700 m/day [31]. One of the main limitations during calibration was related to the spatial coverage of data available for calibration. Future efforts by the BWA should be aimed at establishing a monitoring network that provides more robust data for calibration with fewer gaps. The model could then be re-calibrated and provide a more quantitatively sound description of the island's hydrogeology. A denser network of monitoring locations would also allow for a pilot point calibration approach that bridges the gap between parameter values in every cell of a model and subdivision of models into homogeneous zones [32]. Stochastic techniques could also be investigated to provide probability distribution functions of the hydraulic conductivity field rather than deterministic values.

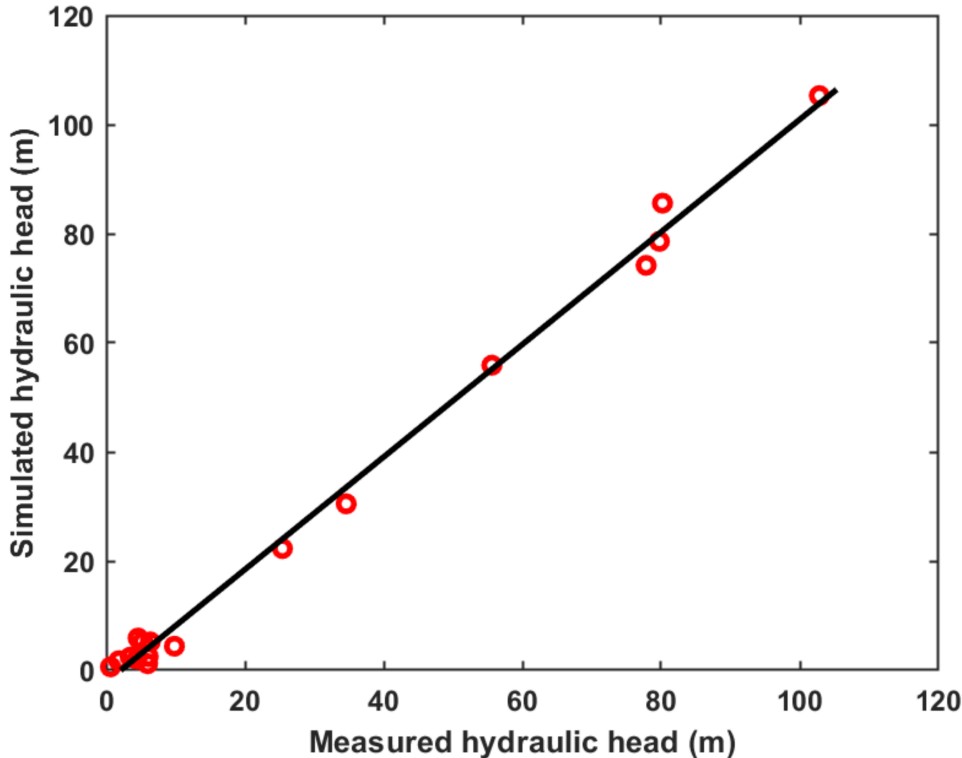

**Figure 6.** Comparison between simulated and observed steady-state hydraulic head after FEFLOW model calibration.

Figure 7 shows regional groundwater flow paths predicted by the physics-based groundwater model. In the north of the island, where there is a groundwater divide owing to an anticlinal limestone geologic feature, the flow paths are directed in opposite directions away from the anticlinal axis. These results provide further support for the model being capable of quantitatively representing Barbados' hydrogeology. The figure also shows the location of sinkholes in reference to the predicted groundwater flow paths. As Barbados moves toward a policy that is aimed at protecting groundwater from contamination, having the quantitative tool developed herein provides part of the basis for risk management. Geophysics surveys that image the subsurface to characterize karst features and prefer-

ential flow paths could be integrated into the developed model, which would support an improved understanding of risks.

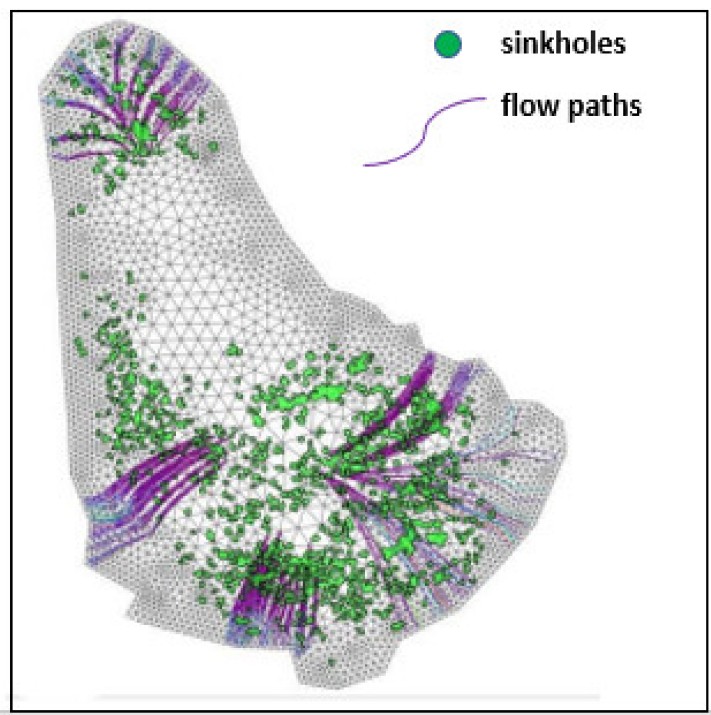

**Figure 7.** Regional groundwater flow paths (purple lines) predicted by the model. The filled green polygons are the locations of sinkholes across the island.

*3.2. Performance of DNNs for Surrogate Modelling*

A suite of numerical experiments was performed to determine the ability of DNN feedforward networks to learn the underlying physics-based groundwater model as well as to investigate the influence of various DNN architectures on performance metrics. The root mean squared error (RMSE), mean absolute error (MAE), and R-squared were the performance metrics used to assess the quality of the surrogate models developed.

To assess the impact of neural network architecture on performance metrics, the network depth and width were varied. The depth of neural networks was varied by specifying the number of hidden layers as 2, 4, and 6, while the width was varied by setting the number of neurons in each hidden layer to 5, 10, 15, and 20. Figure 8a–d show the correspondence between the top four best performing DNN surrogate models. We observed excellent agreement between the neural network surrogate and physics-based groundwater model. The RMSE error was in the range 2.62–2.83 m, the MAE in the range 2.2–2.44 m, and the R-squared ranging from 0.95–0.96 for the best performing surrogates. The deepest network (six hidden layers) with five neurons in each hidden layer produced the best performance metrics. Although neural networks with a single hidden layer are universal function approximators under reasonable assumptions, deeper networks allow for learning more complex functions more efficiently. This explains the best performance observed in the deepest neural network implemented. An automated machine learning approach could be explored in future research efforts to more efficiently explore hyperparameter optimization that would yield improved performance metrics.

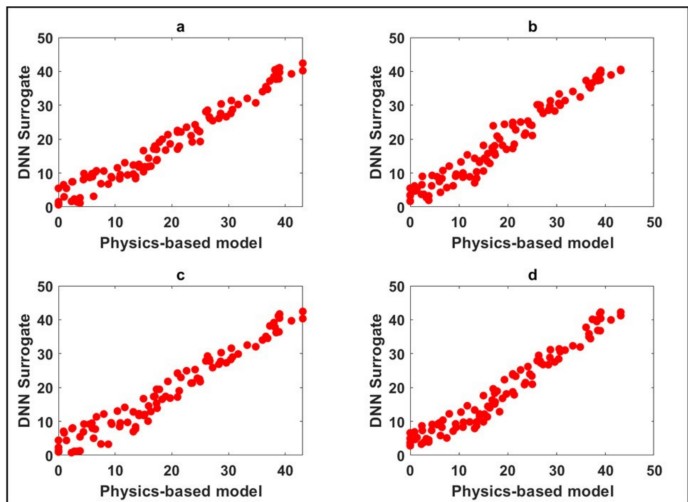

**Figure 8.** DNN surrogate model steady-state hydraulic head (in meters) prediction compared to the physics-based numerical groundwater model steady-state hydraulic head (in meters) estimates. Results from top-4 best performing models are in order of best metrics from (**a**–**d**).

Table 2 summarizes the performance metrics of DNNs with the architectures that produced the best results. DNN(w-x-y-z) indicates w-input features, x-hidden layers, y-neurons in each hidden layer, and z-output features.

**Table 2.** The top performing DNNs with their architecture and corresponding error metrics.

| ML Surrogate | RMSE (m) | MAE (m) | R-Squared |
|---|---|---|---|
| DNN-1 (6-6-5-1) | 2.62 | 2.2 | 0.96 |
| DNN-2 (6-2-15-1) | 2.75 | 2.26 | 0.95 |
| DNN-3 (6-2-20-1) | 2.81 | 2.44 | 0.95 |
| DNN-4 (6-4-10-1) | 2.83 | 2.42 | 0.95 |

### 3.3. Performance of Elastic Nets for Surrogate Modelling

During cross-validation, 10 partitions of the data were generated, the analysis was performed on each partition using elastic nets and an average overall error was determined. Table 3 shows the results for the top performing elastic nets with the corresponding optimal tuning parameters and error metrics (RMSE, MAE, and R-squared). The optimal elastic net was tuned with $\alpha = 1$ and $\lambda = 0.0258$, which led to an RMSE, MAE, and R-squared of 2.74, 2.26, and 0.95, respectively. The performance of elastic nets is similar to DNNs in terms of the ability to approximate the numerical groundwater flow model. The best performing DNN shows a slight improvement over the optimal elastic net in terms of the error metrics used for assessing their performance.

**Table 3.** The top performing elastic nets with their optimally tuned parameters and corresponding error metrics.

| A | λ | RMSE (m) | MAE (m) | R-Squared |
|---|---|---|---|---|
| 0.10 | 0.0258 | 2.74 | 2.25 | 0.95 |
| 0.10 | 0.258 | 2.75 | 2.25 | 0.95 |
| 0.10 | 2.58 | 3.54 | 2.82 | 0.95 |
| 0.55 | 0.0258 | 2.74 | 2.26 | 0.95 |
| 0.55 | 0.258 | 2.76 | 2.27 | 0.95 |
| 0.55 | 2.58 | 3.71 | 2.97 | 0.95 |
| 1.00 | 0.0258 | 2.74 | 2.26 | 0.95 |
| 1.00 | 0.258 | 2.77 | 2.27 | 0.95 |
| 1.00 | 2.58 | 3.87 | 3.11 | 0.95 |

One advantage of elastic nets is the ability to simultaneously perform variable selection and regularization. This was explored by performing a lasso algorithm on the ridge regression coefficients. Variable selection was completed by investigating the coefficients that shrank to zero and keeping the other predictor variables. Table 4 presents the significant coefficients ($p < 0.002$) associated with the predictor variables used in the regression model. Based on the significance of each regression coefficient it was found that recharge had the greatest influence on predicting steady-state hydraulic heads at the specified observation well, while there was a slight influence by well 4. The importance of this result in the context of a stronger scientific basis for groundwater policy is that it can guide optimal well placement, which is an increasingly important consideration as there is an increasing demand and reducing supply due to climate change. The framework developed in this paper can be used to evaluate a suite of pumping and recharge scenarios to evaluate the implications on groundwater availability. However, one caveat is that machine learning algorithms need to be re-trained for new scenarios.

**Table 4.** The significance of each predictor variable coefficient based on the lasso shrinkage model.

| Variable | Significance |
|---|---|
| well 1 | $-3.89 \times 10^{-5}$ |
| well 2 | $6.79 \times 10^{-5}$ |
| well 3 | $1.21 \times 10^{-4}$ |
| well 4 | $1.067 \times 10^{-4}$ |
| well 5 | $-3.32 \times 10^{-4}$ |
| recharge | $4.42 \times 10^{-2}$ |

*3.4. Performance of GANs for Surrogate Modelling*

The training parameters set in the GANs were as follows: the noise dimension was set to 256, the dimension was set to 128, the batch size was set to 32, the log-steps were specified as 100, the learning rate specified was $5 \times 10^{-4}$, and the model was trained for 5000 epochs. The GAN was trained using a generator and discriminator in a zero-sum game to find a model that reproduces the FEFLOW model. The GAN implemented in this paper converged to a result whereby the synthetic data was 95.3% accurate as compared to the physics-based model. In future work we will explore the use of physics-informed neural networks, which will impose the known physics in the loss functions of the neural networks [33,34]. This will lead to less training data being required for training the neural networks.

The numerical groundwater model and machine learning surrogates demonstrate the ability to satisfactorily describe Barbados' hydrogeology. The ML techniques were implemented to approximate steady-state hydraulic heads; however, in the future to better support operational use of these tools a transient response will be approximated. While run times for the physics-based model took approximately 45 min on a Dell Alienware Area 51 R5 system with an 18-core Intel Core i9 7980XE architecture, trained ML surrogates produced similar results on the order of seconds. This is a major advantage of the methodology developed herein from an operational perspective.

Further validation of the model through isotope hydrology studies as well as water level time-series data will be future research thrusts. In addition, the models developed will be coupled to optimization tools for evaluating optimal groundwater management strategies. A limitation of the study is that it investigated the prediction of steady-state hydraulic heads. Predicting a transient groundwater response and variable-density mechanisms would be more challenging and require techniques such as long short-term memory networks. Future work will investigate salinization processes, which represent one of the greatest threats to groundwater quantity and quality on the island and the extension of the current methodology to better capture highly non-linear hydrological processes.

## 4. Conclusions

The objective of this work was to develop a physics-based groundwater model of a small carbonate island and approximate the model using surrogates models based on machine learning techniques. Achieving this objective provides a significant advancement towards data-driven decision making for groundwater resources management under a changing climate. Our results show that a single continuum conceptual model is capable of representing Barbados' hydrogeology at a regional scale. This is supported by the calibration results and groundwater flow patterns predicted by the physics-based numerical groundwater model. Deep neural networks, elastic nets, and generative adversarial networks were developed as surrogates for the physics-based groundwater model and demonstrated their ability to approximate the model with a high degree of accuracy. The approaches developed in this paper provide a path forward for a stronger scientific basis for sustainable groundwater resources management in a small island developing state severely affected by the climate crisis.

**Author Contributions:** Conceptualization, K.P., A.C. and A.M.-J.; Data curation, K.P., I.O., A.M.-J. and J.P.; Formal analysis, K.P., I.O. and P.C.; Funding acquisition, K.P.; Methodology, K.P., A.C., I.O. and A.M.-J.; Project administration, K.P. and A.C.; Resources, K.P., A.C. and D.O.Y.; Software, K.P. and P.C.; Validation, K.P. and P.C.; Visualization, K.P. and I.O.; Writing—original draft, K.P., A.C. and A.M.-J.; Writing—review and editing, K.P., P.C., I.O., D.O.Y., J.P., A.M.-J and A.C. All authors have read and agreed to the published version of the manuscript.

**Funding:** The content is solely the responsibility of the authors and does not necessarily represent the official views of the funding agencies. The authors would like to thank the Global Water Partnership-Caribbean (GWP-C) for funding to support the acquisition of a FEFLOW license and the Microsoft AI for Earth Program for providing computing credits to use the Azure cloud computing service.

**Data Availability Statement:** Water level data were provided by the Barbados Water Authority (BWA).

**Conflicts of Interest:** The authors declare no conflict of interest.

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
