# Peer review of "Machine Learning for Surrogate Groundwater Modelling of a Small Carbonate Island"

_hydrology, doi:10.3390/hydrology10010002_

Round 1

Reviewer 1 Report

The manuscript presents an application of data-driven surrogate models based on deep neural network techniques. There are some interesting elements in the manuscript, the study area is of particular worth and interest to simulate but I see limitations in several parts of the work and therefore my recommendation is for major revisions. Please find below some questions/comments for possible improvement of this manuscript:

1)      Introduction: In general, this part of the manuscript is not well structured and a large proportion of it appears too specific and related to the study area and should be moved there. The title of the manuscript prepares the reader for an introduction which rather discusses the capabilities of Machine Learning (ML) to groundwater modelling applications, a reasonable coverage of the relevant past literature, some connection to small island hydrology in general and what are the novelties of this work. If the intention is publishing a case study with the use of surrogate modelling on top, then the title should be changed accordingly but, in any case, the whole structure of the introduction needs re-writing. Also, since the model application refers to a small island there are already several papers in the literature applying data-driven surrogate models for coastal aquifer flow. Some of those should be at least referenced given that most of those studies apply more complex variable-density flow and solute transport models. It might be the case that the authors should then highlight the fact that part of this paper’s novelty is why their ML approach is interesting to the groundwater modelling community.

2)      Line 94-95: Based on section 2.2 the statement that the present model “accurately represents Barbados’ hydrogeology” to my understanding is an overstatement.

3)      Unfortunately, despite that the numerical model is interesting significant elements are missing, particularly that variable-density flow is neglected. That would have made the ML task much more interesting given the strong non-linearities that need to be captured by the data-driven surrogates.

4)      I couldn’t see any reference to the computational cost of a single runtime of the numerical model so the reader can understand what 500 simulations mean and why 400 samples, which is already a significant amount of runs, was selected for training.

5)      What method was used for preparing the training samples?

6)      If I correctly understand, the ML techniques emulate steady-state heads as model output. This is a bit of disappointment, emulating the transient response would have been far more interesting and challenging.  

7)      Based on the very clustered monitoring data of figure 4, my understanding is that assumptions that the present model accurately represents the hydrogeology of the area are not valid and this also narrows down the challenge of the ML surrogates to show how they perform spatially.

8)      In overall, the discussion of challenges and limitations of this work is not adequate, and I think much more effort should be put to the manuscript to make the content publishable and of international interest.

Author Response

Please see the responses to comments attached.

Reviewer 2 Report

This manuscript, hydrology-2034886-peer-review-v1- entitled "Machine Learning for Surrogate Groundwater Modelling of a Small Carbonate Island," is well written and has potential, but it should be more organized.

In my opinion, a careful revision of the English language should be carried out as there currently are some unclear sentences. The study seems to be well designed. The methodology and results are technically sound. Discussions on the scientific and practical values of the study, the limitations of proposed models, and future work are meaningful. I recommend accepting this manuscript after revision. The main concerns are as follows:

1)     Quantitative results should be provided in the abstract to make it more comprehensive. Results of ML should be added in the abstract section. Also, The main aim of the study should be clearly mentioned in the abstract.

2)     More recent references might support the first and second paragraphs of the introduction. Some references and literature are pretty old. There is two research reference for 2021 and 2022. The authors should read and use the newly published papers in their research.

3)     More literature review about the other methods is needed. The manuscript could be substantially improved by relying and citing more on recent literature about contemporary real-life case studies of sustainability and/or uncertainty, such as the following

·       Vadiati, M., Rajabi Yami, Z., Eskandari, E., Nakhaei, M., & Kisi, O. (2022). Application of artificial intelligence models for prediction of groundwater level fluctuations: Case study (Tehran-Karaj alluvial aquifer). Environmental Monitoring and Assessment, 194(9), 1-21.

4)     For readers to quickly catch your contribution, it would be better to highlight significant difficulties and challenges and your original achievements to overcome them more straightforwardly in the abstract and introduction.

5)     Providing a comprehensive flowchart is highly recommended by researchers, so please add a flowchart representing the methodology in the paper.

6)     Island of Barbadosis adopted as the case study. What are other feasible alternatives? What are the advantages of adopting this case study over others in this case? How will this affect the results? The authors should provide more details on this.

7)     It is important to give a better description of the samples and the sampling protocol since we are trying to understand the data variability. What are the advantages of adopting these parameters over others in this case? How will this affect the results? More details should be furnished.

8)     It is better to add more error criteria to better understand the model's ability.

9)     The discussion section in the present form is relatively weak and should be strengthened with more details and justifications.

10)  It seems that conclusions are observations only, and the manuscript needs thorough checking for explanations given for results. The authors should interpret more precisely the results argument.

Author Response

Please see the response to the article comments attached.

Reviewer 3 Report

Developing surrogate models to emulate intensively computational models is justified by their use. While the authors have prepared a vey nice and very well written paper, there was no discussion of this important question. It is quite possible that the numerical model is impractical when doing stochastic studies including climate change scenarios. There were also a few weak points in the text that could easily be addressed and would make a much stronger paper. Below are some other comments that need to be addressed. I believe the authors have all the information needed to answer the questions above, and the comments/editorials below for a quick re-write of the paper.

Line 27: “…digital twin…”. Did authors mean “…digital tool…”?

Line 146: Text refers to “sinkholes”. Sinkholes may mean different things depending on context. A clarification would be very helpful.

Lines 157: Figure 2 is very lacking in information for a general reader: in geographic reference, scale, North Arrow, etc. A GIS-based figure would be very helpful.

Lines 171-197: Detailed description of the numerical model is absent. What is the model area, grid, grid size, transient/steady state, time step, layers, simulation/verification periods, calibration/verification results, etc.?

Line 214-215: Were the training/testing samples chosen randomly?

Lines 229, 245, 253: variables in Equations 5, 6, 7, and 8 not clearly defined.

Line 271: The title to Figure 5 does not match the content of the figure itself. Also, in Figure 5, data points are shown with no context!

Lines 287 through 366: In describing the surrogate models, there was no discussion of testing/verification/validation of the machine learning techniques. Also, what is the savings in run time for the original numerical model vs. surrogate model?

Line 310: In Figure 7, there are no units.

Line 317: In Table 2 there is no reference to compare the statistical results to. i.e, there is no discussion of the observations themselves, to compare predictions to.

Line 434: Reference #16 was not cited in the main text.

Author Response

Please see the attached for responses the the reviewer's comments.

Reviewer 4 Report

Attempting to model a karstic carbonate island with a neural network model seems to have worked - better than I would have expected.  I have only a few comments and corrections.

The study area could be described better.  I'm not sure what is meant by t he term "streamwater" on line119.  Conduit flow in a karstic aquifer?  The karstic character of the aquifer is described only in terms of sinkholes.  What is known about the subsurface?  Major caves?  Large springs?  Is all of the aquifer discharge below sea level?  In general, what is known about the subsurface properties of the aquifer?

Figure 5 has an incorrect caption.

What is being plotted in Figure 7?  The figures compare results from the physics based model with DMN  but what is the actual parameter being plotted?

Author Response

Please see the attached for responses to the article comments.

Round 2

Reviewer 1 Report

The authors generally responded adequately to my comments. I hope that in a future study they will incorporate variable-density flow and/or transient simulations in order to better represent the hydrological challenges. Please have a look again at lines 409-426 in the revised manuscript, the text needs revising for spelling and content errors

Author Response

Thanks again for your feedback and we will be undertaking a more complex approximation of variable-density and transient responses.

We have reviewed lines 409-426 and removed repetitive language and checked for typos.

Reviewer 3 Report

Thank you for addressing my comments. I accept the modifications. Well done.

Author Response

The authors would like to thank the reviewer for their insightful and useful feedback, which helped to improve the manuscript.